# Implementation of the Simple Hyperchaotic Memristor Circuit with Attractor Evolution and Large-Scale Parameter Permission

**DOI:** 10.3390/e25020203

**Published:** 2023-01-19

**Authors:** Gang Yang, Xiaohong Zhang, Ata Jahangir Moshayedi

**Affiliations:** 1School of Information Engineering, Jiangxi University of Science and Technology, Ganzhou 341000, China; 2Khomeini Shahr Branch, Islamic Azad University, Isfahan 10587, Iran

**Keywords:** hyperchaotic, memristor, attractor evolution, large-scale parameter permission, spectral entropy complexity, coexisting attractors, FPGA

## Abstract

A novel, simple, four-dimensional hyperchaotic memristor circuit consisting of two capacitors, an inductor and a magnetically controlled memristor is designed. Three parameters (*a*, *b*, *c*) are especially set as the research objects of the model through numerical simulation. It is found that the circuit not only exhibits a rich attractor evolution phenomenon, but also has large-scale parameter permission. At the same time, the spectral entropy complexity of the circuit is analyzed, and it is confirmed that the circuit contains a significant amount of dynamical behavior. By setting the internal parameters of the circuit to remain constant, a number of coexisting attractors are found under symmetric initial conditions. Then, the results of the attractor basin further confirm the coexisting attractor behavior and multiple stability. Finally, the simple memristor chaotic circuit is designed by the time-domain method with FPGA technology and the experimental results have the same phase trajectory as the numerical calculation results. Hyperchaos and broad parameter selection mean that the simple memristor model has more complex dynamic behavior, which can be widely used in the future, in areas such as secure communication, intelligent control and memory storage.

## 1. Introduction

The memristor describes the relationship between magnetic flux and charge, which was proposed by Chua in 1971 based on the theory of electronics [1,2]. The memristor has intrinsic properties, and it is considered as the fourth basic circuit element, in addition to the capacitor, inductor and resistor [3]. In 2008, the team of Stanley Williams of the Hewlett-Packard Lab in the United States successfully developed a solid-state memristor based on Chua’s memristor idea [4]. Hence, the memristor has attracted extensive attention and has been studied in depth. Since its introduction, the magnetic-controlled memristor [5], the generalized magnetic-controlled memristor [6], the smooth quadratic nonlinear memristor model [7], the magnetic-controlled memristor model with absolute value and square root algorithms [8], the multistable memristor [9], etc., have been proposed successively. The characteristics of hyperchaotic systems depend on the number of positive Lyapunov exponents [10,11], while nonlinear systems combine the memory and nonlinear properties of memristors to generate chaotic behaviors. Considering the complexity of the circuit structure, many memristor circuits are difficult to design and implement. Therefore, the design of simple memristor circuits has also attracted attention. In recent years, the simple chaotic circuit based on memristors [12,13,14], the simple chaotic circuit based on a memristor and capacitor [15,16], the simple parallel chaotic circuit based on a memristor [17], the simple double-scroll chaotic circuit based on a meminductor [18] and many other simple memristor chaotic circuits have been studied. Thus far, a variety of memristor-based hyperchaotic circuits have been proposed—for example, the Wien-bridge hyperchaotic memristive circuit system [19], the novel memristor-based symmetric hyperchaotic circuit system [20], the hyperchaotic circuit system based on memristor feedback with multistability and symmetries [21], the hidden extreme multistability in memristive hyperchaotic system [22], the hyperchaotic memristive system with hidden attractors [23], the no-equilibrium memristive system with a four-wing hyperchaotic attractor [24], etc. Combining the characteristics of a memristor, a simple chaotic system and a hyperchaotic system, a simple hyperchaotic memristor circuit is proposed in this paper.

The traditional methods for analyzing chaotic systems include the Lyapunov spectrum, bifurcation diagram, equilibrium point [25,26], etc., and there are spectral entropy complexity methods [27,28,29] for the research methods of chaotic systems with complex nonlinear dynamic behavior. Chaotic systems are extremely sensitive to initial conditions [30], and some chaotic systems are also extremely sensitive to parameter changes, which leads to the phenomenon of attractor evolution [16]. In addition, there are extended studies performed by changing the initial conditions of chaotic systems to generate coexisting attractors

[15,31,32,33,34]. Meanwhile, the multistability of a chaotic system depends on the initial conditions of the system, and the basin of attraction is a method to analyze the multistability of a chaotic system. The dynamic behavior of the system can be well distinguished according to the different attraction regions of the basin of attraction [31,34,35,36,37]. With the in-depth study of memristor chaotic circuits, analog memristor circuits are easily affected by the external environment, while FPGA technology has the advantages of high flexibility, parallel computing, low power consumption and high reliability [38,39]. Therefore, the design of memristor chaotic circuits using FPGA is significant for practical applications. Nowadays, a variety of memristor chaotic systems have been implemented by FPGA [34,40,41,42,43]—for instance, FPGA realization of a novel 5D hyperchaotic four-wing memristive system [44], FPGA implementation of a memristor-based multi-scroll hyperchaotic system [45], etc.

The significant contributions of this paper are as follows:

(1) A novel, simple, four-dimensional memristor chaotic circuit is designed, whose structure consists of a magnetically controlled memristor, two capacitors and an inductor. The circuit is a hyperchaotic system and exhibits complex dynamics.

(2) The simple memristor chaotic circuit is extremely sensitive to changes in the internal parameters *a*, *b* and *c*, and each parameter change can cause the circuit to evolve a large number of different types of chaotic attractors.

(3) By numerical simulation, it is found that the parameters *a*, *b* and *c* of the circuit have large-scale parameter permission. Meanwhile, the spectral entropy complexity of the circuit and dynamic performance are verified in detail.

(4) The simple memristor chaotic circuit is realized by FPGA technology, and the results of the hardware experiment are consistent with the numerical simulation, which proves that the design is feasible and correct.

In this paper, the magnetically controlled memristor model and its characteristic curve are introduced, and the simple memristor chaotic circuit is composed by combination with a small number of circuit elements. The circuit has two positive Lyapunov exponents, so it is a hyperchaotic system. The circuit is very sensitive to parameter changes; not only does its chaotic attractors evolve, but also the internal parameters of the circuit have large-scale range capabilities. The chaotic properties of the simple memristor chaotic circuits are investigated by using spectral entropy complexity, coexisting attractors and the basin of attraction method. In addition, the circuit is implemented by FPGA technology.

The paper is organized as follows. In Section 2, the magnetically controlled memristor model and its characteristic curves are introduced, and the simple four-dimensional memristor chaotic circuit is proposed and analyzed. In Section 3, the attractor evolution phenomenon of the circuit, the influence of large-scale parameters, spectral entropy complexity analysis, coexisting attractor behavior and the study of the basin of attraction are described in detail. In Section 4, the simple memristor chaotic circuit is designed with FPGA technology. The paper is summarized in Section 5.

## 2. Analysis of the Simple Memristor Chaotic Circuit

### 2.1. The Structure of the Magnetically Controlled Memristor Model

According to the basic definition of memristors, they can be classified into magnetic-controlled memristors and charge-controlled memristors. A new type of magnetic-controlled memristor is proposed, and its expression is as follows:(1)$$\left\{\begin{array}{c} i=\alpha \varphi v\\ \dot{\varphi }=\beta v^{2}+\gamma v+\delta \end{array}\right.$$
where α, β, γ and δ are parameters, $\dot{\varphi }$ is the internal variable of the magnetic-controlled memristor, *i* is the current variable and *v* is the voltage variable.

To confirm that model (1) is a memristor, we set α = 1, β = 1, γ = 0.5, δ = −3 and the input voltage is *v* = *A*sin(2π*ft*). When the amplitude is *A* = 3 V, the characteristic curves of the magnetically controlled memristor are as shown in Figure 1a under the excitation of frequency *f* = 1 Hz, *f* = 2 Hz and *f* = 5 Hz, respectively. The effect of amplitude *A* = 4 V, *A* = 3.5 V and *A* = 3 V on the magnetically controlled memristor at a constant frequency *f* = 1 Hz is shown in Figure 1b. It can be seen that the characteristic curves all pass through the origin in the *v*-*i* plane from Figure 1. Meanwhile, Figure 1a shows that as the frequency *f* increases, the area of the tight hysteresis loop decreases. The increase in amplitude *A* causes the hysteresis line loop to increase as well in Figure 1b. These features verify the memristor characteristic of model (1) [3].

### 2.2. Design of the Simple Memristor Chaotic Circuit

The simple memristor chaotic circuit consists of two capacitors $C_{1}$, $C_{2}$, an inductor *L* and a magnetically controlled memristor W(φ), as shown in Figure 2.

According to Kirchhoff’s theorem and the relationship between circuit elements, the dynamic behavior of the circuit is described as follows:(2)$$\begin{array}{c} \left\{\begin{array}{c} \;\frac{dV_{1}}{dt}=-\frac{1}{C_{1}}i_{L}\\ \frac{dV_{2}}{dt}=\frac{1}{C_{2}}\left(i_{L}-\alpha \varphi V_{2}\right)\\ \frac{di_{L}}{dt}=\frac{1}{L}\left(V_{1}-V_{2}\right)\\ \frac{d\varphi }{dt}=\beta {V_{2}}^{2}+\gamma V_{2}+\delta \end{array}\right. \end{array}$$
where α, β, γ and δ are parameters, $C_{1}$, $C_{2}$ are capacitance values, and *L* is the inductance value.

We set parameters *a* = 1/$C_{1}$ = 8, *b* = 1/$C_{2}$ = 1, *c* = 1/*L* = 2, α = 1, β = 1, γ = 0.5, δ = −3 in system (2). After selecting x(t) = $V_{1}\left(t\right)$, y(t) = $V_{2}\left(t\right)$, z(t) = $i_{L}\left(t\right)$, w(t) = φ(t), the expression of system (2) becomes the four-dimensional state Equation (Equation 3):(3)$$\begin{array}{c} \left\{\begin{array}{c} \dot{x}=-az\\ \dot{y}=b\left(z-yw\right)\\ \dot{z}=c\left(x-y\right)\\ \dot{w}=y^{2}+0.5y-3 \end{array}\right. \end{array}$$

Setting the initial conditions of system (3) as $\left(x_{0},y_{0},z_{0},w_{0}\right)$ = (0.1, 0.1, 0.1, 0.1), the state trajectory diagram of system (3) is as shown in Figure 3. The multi-scroll chaotic attractor corresponding to the different variables can be observed. To verify the chaotic properties of system (3), we used the classical Wolf algorithm to calculate the Lyapunov exponents [10]; the corresponding Lyapunov exponents are calculated as $LE_{1}$ = 0.0433, $LE_{2}$ = 0.0159, $LE_{3}$ = 0, $LE_{4}$ = −0.0521, where there are $LE_{1}$, $LE_{2}$ greater than zero, $LE_{3}$ equal to zero and $LE_{4}$ less than zero, and the nature of the Lyapunov exponents is (+, +, 0, −). Thus, system (3) is a hyperchaotic system [11].

## 3. Nonlinear Dynamics Analysis of the Simple Memristor Chaotic Circuit

### 3.1. Phenomenon of Attractors’ Evolution

Generally, when the internal parameters of a chaotic system change, the nonlinear behavior of the system will also be affected by it. Through a large number of numerical simulations, it is found that system (3) is extremely sensitive to changes in the internal parameters, and the change in parameters will also cause the state trajectory variation of system (3).

Experiment 1: parameter a change. Changing only the value of the parameter *a*, keeping the other parameters constant and the initial condition of $\left(x_{0},y_{0},z_{0},w_{0}\right)$ = (0.1, 0.1, 0.1, 0.1), the chaotic attractors of *y*-*z* corresponding to different parameters *a* are as shown in Figure 4 and marked in red. It can be observed from Figure 4 that different values of the parameter *a* correspond to different types of attractors. We found that a great deal of different attractors are classified in the range of a∈(0,200] and we selected 9 types from them, so we defined these attractors as type-1 to type-9, respectively. The Lyapunov exponent spectrum and bifurcation diagram with the variation of the parameter *a* are shown in Figure 5. Meanwhile, Table 1 summarizes in detail the Lyapunov exponents, dynamic state and Lyapunov dimensions of the attractors corresponding to the values of the fixed parameter *a*. According to the value range of a∈(0,200], each similar type of chaotic attractor is classified in detail and can be seen in Table 1.

Experiment 2: parameter b change. Similarly, we set the parameters of system (3) to be consistent and the initial conditions to $\left(x_{0},y_{0},z_{0},w_{0}\right)$ = (0.1, 0.1, 0.1, 0.1). System (3) also generates a great number of chaotic attractors of *y*-*z* by varying only the parameter *b* in the range b∈(0,60], as shown in Figure 6 and marked in blue. At the same time, Figure 7 shows the corresponding Lyapunov exponent spectrum and bifurcation diagram, and the detailed data can be found in Table 2.

Experiment 3: parameter c change. The parameters and initial conditions of system (3) are also consistent with the above two experiments. When the parameter *c* is varied within (0, 11.15], Figure 8 shows that system (3) evolves 9 different *y*-*z* attractors, marked in green. Moreover, the Lyapunov exponent spectrum and bifurcation diagram with parameter *c* are shown in Figure 9. Table 3 indicates the relevant properties of each type of attractor.

Comparing the parameter *a* and the parameter *b*, it is found that the parameter *c* varies within (0, 11.15], the attractor evolution behavior of system (3) is particularly prominent, and the behavior in a large range is extremely weak. Obviously, the orbit of the circuit is different from the general chaotic system, and the attractor evolution behavior caused by the internal parameters not only is sensitive to the changes in parameters *a*, *b* and *c*, but also illustrates the rich dynamical characteristics of system (3). The randomness and irregularity of this property provide a good application in the field of information security; they can form a large capacity of key space, as pseudo-random signals are very difficult to crack and recognize.

### 3.2. Wide Range of Chaotic Characteristics of the Internal Parameters
of the Circuit

The chaotic system maintains complex chaotic behavior only when the parameters are varied within a certain range. However, when the parameters are changed in a wide range, system (3) in this paper still maintains a chaotic state. In the following, one of the three parameters (*a*, *b*, *c*) is changed separately to show the advantages of large-scale parameters in this paper.

For instance, setting the initial conditions as $\left(x_{0},y_{0},z_{0},w_{0}\right)$ = (0.1, 0.1, 0.1, 0.1), the Lyapunov exponents spectrum and the bifurcation diagram with the change in parameter *a* are as shown in Figure 10a,b. Obviously, it can be seen from Figure 10a that the parameter *a* has a positive Lyapunov exponent from 0 to 1000. Furthermore, the corresponding bifurcation diagram in Figure 10b maintains the clustering phenomenon of dense points. This illustrates that system (3) always maintains a chaotic state when the parameter *a* is (0, 1000]. For example, when the parameter *a* = 1000, the corresponding Lyapunov exponents are $LE_{1}$ = 0.0165, $LE_{2}$ = 0, $LE_{3}$ = 0 and $LE_{4}$ = −0.0167. At the same time, we also test system (3) on a larger scale, and system (3) still has chaotic behavior.

In addition, the parameter *b* and parameter *c* were also investigated separately in a large range, and it was discovered that system (3) also always maintains the chaotic state, as shown in Figure 11 and Figure 12. It can also be deduced that both the parameters *b* and *c* vary in a wide range of (0, 1000]. In fact, the parameters *b* and *c* also have a larger range, similar to the parameter *a*, which causes system (3) to continue to be chaotic. The Lyapunov exponent spectrum and the corresponding bifurcation diagram in Figure 10, Figure 11 and Figure 12 not only verify that system (3) has the performance of a persistent chaotic state, but also has very complex nonlinear dynamic behavior. Moreover, the parameters *a*, *b* and *c* have a wide range of chaotic properties, as well as the attractor evolution behavior described in Section 3.1. The chaotic random sequence generated by system (3) has strong pseudorandomness over a wide range.

### 3.3. Spectral Entropy Complexity Analysis of the Simple Memristor
Chaotic Circuit

Complexity is an important method for analyzing chaotic dynamical systems, where the spectral entropy (SE) and $C_{0}$ algorithms are suitable choices for the accurate estimation of time series complexity. SE complexity is a spectral entropy value obtained by using the Fourier transform combined with Shannon entropy. $C_{0}$ complexity is a spectral entropy value obtained by using the FFT transform to remove the regular spectrum in the signal transform domain while leaving the irregular spectrum entropy [27,28,29].

Given a pseudorandom sequence x(n),n=0,1,2,⋯,N−1 of length *N*, such that $x\left(n\right)=x\left(n\right)-\bar{x}$, where $\bar{x}$ is the mean of the time series, its corresponding discrete Fourier transform is defined as
(4)$$X\left(k\right)=\sum_{n=0}^{N-1}x\left(n\right)e^{-j2\pi nk/N}$$
where k=0,1,2,⋯,N−1.

The relative power spectrum probability of this sequence can be expressed as
(5)$$P_{k}=\frac{{\left|X\left(k\right)\right|}^{2}}{\sum_{n=0}^{N/2-1}{\left|X\left(k\right)\right|}^{2}}$$

Based on the concept of Shannon entropy and the value of spectral entropy converging to ln(N/2), the normalized spectral entropy SE can be expressed as
(6)$$SE\left(N\right)=\frac{-\sum_{k=0}^{N/2-1}P_{k}\ln P_{k}}{\ln\left(N/2\right)}$$

Removing the irregular part of Equation (Equation 4), the parameter *r* is introduced to keep the spectrum that is more than *r* times the mean square value, and we set the rest to 0, which is
(7)$$\tilde{x}\left(k\right)=\left\{\begin{array}{c} X\left(k\right),\;\;{\left|X\left(k\right)\right|}^{2}>rG_{N}\\ 0,\;\;{\left|X\left(k\right)\right|}^{2}\leq rG_{N} \end{array}\right.$$
where $G_{N}$ is the mean square value of the sequence.

The expression for the Fourier inverse transform of Equation (Equation 7) is
(8)$$\tilde{x}\left(n\right)=\frac{1}{N}\sum_{k=0}^{N-1}\tilde{X}\left(k\right)e^{j2\pi nk/N}$$

From the above, the $C_{0}$ complexity can be defined as
(9)$$C_{0}\left(r,N\right)=\sum_{n=0}^{N-1}{\left|x\left(n\right)-\tilde{x}\left(n\right)\right|}^{2}/\sum_{n=0}^{N-1}{\left|x\left(n\right)\right|}^{2}$$

Setting the initial conditions of the system (3) as $\left(x_{0},y_{0},z_{0},w_{0}\right)$ = (0.1, 0.1, 0.1, 0.1), the SE complexity and $C_{0}$ complexity as the parameter *a* varies are as shown in Figure 13, and Figure 14 shows the SE and $C_{0}$ complexity as the parameter *b* varies. According to the information shown in Figure 13 and Figure 14, it can be seen that the fluctuations of SE complexity and $C_{0}$ complexity are similar. Moreover, comparing Figure 13 with Figure 10b and Figure 14 with Figure 11b, it can be found that the trend of the two complexities is also largely consistent with the total trend of the bifurcation diagram. It can be roughly inferred from Figure 13 that the normalized average value of SE is 0.23 and the average value of $C_{0}$ spectral entropy is 0.03, while the normalized average value of SE is approximately 0.57 and the average value of $C_{0}$ spectral entropy is approximately 0.18 in Figure 14. Apparently, the results of Figure 13 and Figure 14 show that the normalized average value of SE and the average value of $C_{0}$ spectral entropy corresponding to parameter *b* are greater than that of parameter *a*. This not only indicates that Figure 14 contains more complex information, but also that the variation of parameter *b* has a greater impact on system (3).

The Lyapunov dimension is a quantitative measure of the complexity of chaotic systems [46], so different values of parameters *a* and *b* were chosen to calculate the corresponding Lyapunov dimensions, which can be seen in Table 4. It can be observed from Table 4 that the calculated four Lyapunov dimension values are all around 4, which indicates that system (3) is a very complex four-dimensional chaotic system.

It has been verified in Section 3.2 that large changes in the parameter *a* and the parameter *b* can keep the system (3) in the chaotic state. Let the parameters *a* and *b* be changed to obtain the corresponding complexity, as shown in Figure 15. It can be judged from Figure 15 that the black region is the highest complexity value, followed by the red region, and the yellow region is the lowest. The larger complexity value indicates that the sequence is closer to the random sequence and the security of the corresponding system is higher. It can be deduced from Figure 15 that the complexity is in the black and red regions when the parameters *a* and *b* change considerably, which shows that system (3) contains very complex information and the system has good security.

### 3.4. Coexistence of Attractors

Multistability is a unique phenomenon in nonlinear systems, and coexisting attractors have the property of multistability. Generally, coexisting attractors are classified as symmetric coexisting attractors and asymmetric coexisting attractors, where symmetric coexisting attractors are characterized by attractors belonging to the same type and symmetric about the coordinate axis or origin [31,32,33]. The proposed system (3) in this paper has multiple symmetric coexisting attractors.

The parameters of system (3) remain unchanged. Using the initial conditions $\left(x_{0},y_{0},z_{0},w_{0}\right)$ = (0.1, 0.1, 0.1, 0.1) as a reference, we only change any two of the initial values. For example, a set of initial condition values is chosen as $\left(x_{0},y_{0},z_{0},w_{0}\right)$ = (0.1, −4, 0.1, 0.01/−0.01), where only changes to $y_{0}$, $w_{0}$ are made, while $x_{0}$ and $z_{0}$ remain the same, corresponding to the coexisting attractors shown in Figure 16. To better reflect the coexistence characteristics of the coexisting attractors, Figure 17 and Figure 18 show the time-domain waveforms of the variables *x*, *y* and the bifurcation diagram of the variation of the parameter *b*. It can be observed from Figure 17 that the *x*, *y* time-domain waveform traces of the initial value (0.1, −4, 0.1, 0.01) and (0.1, −4, 0.1, −0.01) are consistent. Similarly, it can be found in Figure 18 that the corresponding bifurcation diagrams have essentially the same trend in direction. In addition, other coexisting attractors with varying initial values are shown in Figure 19, and detailed data can be obtained from Table 5. It can be discovered that each set of initial conditions in Figure 19 corresponds to a completely different coexisting attractor. These phenomena of coexisting attractors illustrate that system (3) is not only multistable but also has complex dynamical behavior.

### 3.5. Analysis of Attraction Basins

Basin of attraction is an analytical method for studying the dynamical behavior of chaotic systems, and it can discriminate the types of attractors in which chaotic systems exhibit bounded behavior. Moreover, generating different regions of attraction indicates the existence of coexisting multistability [35,36,37]. By changing the initial values of any two variables within a certain range, while the initial values of other variables remain unchanged, the corresponding basins of attraction are obtained, as shown in Figure 20.

The red region is the attraction region of the periodic chaotic attractors, and the yellow region is also the attraction region of the chaotic attractors, which implies the coexistence behavior of multiple attractors. The coexisting attractors of Table 5 in Section 3.4 are almost in the red attraction region and are all marked in Figure 20. The regions in red and yellow also represent completely different initial conditions, which shows the coexistence of multiple stabilization phenomena and the existence of symmetry in each of the attractor basins in Figure 20. The corresponding initial conditions $S_{1}$–$S_{12}$ are also identified in Figure 20, which are consistent with the results in Figure 19. Therefore, system (3) has multiple attractor coexistence behavior.

## 4. FPGA Implementation of the Simple Memristor Chaotic Circuit

FPGA technology has the advantages of high flexibility and parallel computing [38,39], so a hardware experiment was performed on the simple memristor chaotic circuit (3) based on FPGA. The main platform of the experiment adopts a Cyclone IV E series, FPGA main chip with model EP4CE10F17C8N, 14-bit dual-channel AD9767 DAC chip and oscilloscope. The experimental hardware platform is shown in Figure 21, where the PLL module configures the appropriate frequency for the digital chaotic system module.

### 4.1. Discretization of Simple Memristor Chaotic Circuit Model

We use Verilog HDL programming and signed fixed points to design the circuit model (3) accordingly. Because model (3) is extremely sensitive to changes in parameters, the fixed-point format is 1-bit signal bit, 5-bit integer bit and 26-bit decimal bit. At the same time, considering the accuracy problem and the relative simplicity of model (3), the improved Euler algorithm is used to discretize model (3); we call this a decomposition of the time domain. The discretized simple memristor chaotic circuit can be described as follows:(10)$$\left\{\begin{array}{c} x\left(n\right)=\bar{x}\left(n\right)+\Delta t\left[-a\bar{z}\left(n\right)\right]\\ y\left(n\right)=\bar{y}\left(n\right)+\Delta t\left[b\bar{z}\left(n\right)-b\bar{y}\left(n\right)\bar{w}\left(n\right)\right]\\ z\left(n\right)=\bar{z}\left(n\right)+\Delta t\left[c\bar{x}\left(n\right)-c\bar{y}\left(n\right)\right]\\ w\left(n\right)=\bar{w}\left(n\right)+\Delta t\left[{\bar{y}}^{2}\left(n\right)+0.5\bar{y}\left(n\right)-3\right] \end{array}\right.$$
(11)$$\left\{\begin{array}{c} \bar{x}\left(n+1\right)=\bar{x}\left(n\right)+\frac{\Delta t}{2}\left[x\left(n\right)+x\left(n+1\right)\right]\\ \bar{y}\left(n+1\right)=\bar{y}\left(n\right)+\frac{\Delta t}{2}\left[y\left(n\right)+y\left(n+1\right)\right]\\ \bar{z}\left(n+1\right)=\bar{z}\left(n\right)+\frac{\Delta t}{2}\left[z\left(n\right)+z\left(n+1\right)\right]\\ \bar{w}\left(n+1\right)=\bar{w}\left(n\right)+\frac{\Delta t}{2}\left[w\left(n\right)+w\left(n+1\right)\right] \end{array}\right.$$
where $\Delta t=2^{-11}$ is the time sampling step. Equation (Equation 10) is the expression of Euler’s algorithm for the circuit model (3).

### 4.2. State Machine Execution Operation

State machine programming has the advantages of logical completeness, a clear program structure and flexibility, so the state machine is used to design system (11). The state machine contains five states and the detailed process is described below:Initialize the initial conditions $\left(x_{0},y_{0},z_{0},w_{0}\right)$ of the circuit;Intercept the calculated 64-bit yw and yy;Parallel calculation to obtain the result of x(n), y(n), z(n) and w(n);Repeat the above steps to calculate x(n+1), y(n+1), z(n+1) and w(n+1);According to Equation (Equation 11), add and shift to obtain the final result $\bar{x}\left(n+1\right)$, $\bar{y}\left(n+1\right)$, $\bar{z}\left(n+1\right)$ and $\bar{w}\left(n+1\right)$.

### 4.3. FPGA Implementation and Comparison

The 14-bit output data are intercepted and burned into the EP4CE10F17C8N, which is then converted into an oscilloscope via the AD9767, and the experimental platform is connected, as shown in Figure 22. Finally, the chaotic attractors of the simple memristor chaotic circuit (3) shown on the oscilloscope are as shown in Figure 23.

Comparing the state trajectories of the attractors of type-2, type-4 and type-5 in Figure 4; the attractors of type-3, type-5 and type-6 in Figure 6; and the attractors of type-2, type-3 and type-4 in Figure 8, the results of both are essentially the same. The experimental results of FPGA are accurate and consistent with the results of the MATLAB simulation.

## 5. Conclusions

In this paper, the characteristic curves of the magnetically controlled memristor model are first analyzed. Then, a novel four-dimensional simple memristor chaotic circuit is proposed by using a magnetically controlled memristor in combination with other circuit elements, and the circuit is analyzed as a complex hyperchaotic system from the theoretical results. The internal parameters *a*, *b* and *c* of the circuit are tested separately, and it is deduced that the circuit is extremely sensitive to parameter changes. Moreover, the circuit can generate a large number of different types of chaotic attractors under the influence of the parameters *a*, *b* and *c*, respectively. At the same time, the circuit maintains the chaotic characteristics for each parameter (*a*, *b* or *c*) varying in a large-scale range of (0, 1000]. By investigating the spectral entropy complexity of the circuit, it is further illustrated that the circuit contains a large amount of nonlinear dynamical behavior. Next, multiple sets of symmetric initial conditions are chosen, and the circuit has many different coexisting attractors, while the internal parameters of the circuit are kept consistent. In addition, setting any two initial values of the range and keeping the other two constant, the circuit’s basin of attraction is classified into two chaotic attractors with red and yellow attraction regions, which reflects the multistability and dynamic characteristics of the circuit. Finally, based on the time-domain method, a simple memristor chaotic circuit is implemented by FPGA, and the experimental results are compared with the numerical simulation results to verify the accuracy of the experimental results. Therefore, the circuit has wide potential for application in secure communications, neural networks and engineering applications. In this paper, the complex dynamic behaviors studied are based on integer-order settings, and future research can be extended in the field of fractional order.

## Figures and Tables

**Figure 1 entropy-25-00203-f001:**
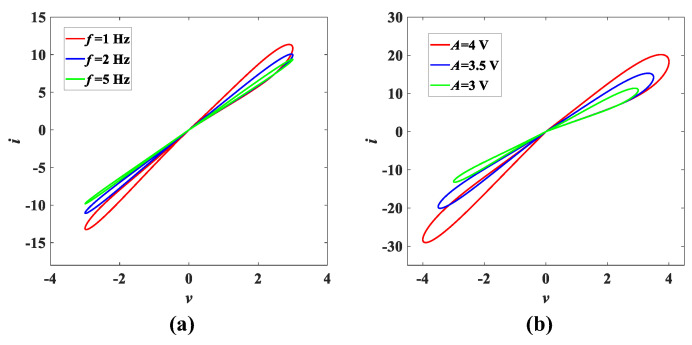
The *v*-*i* characteristic curves of magnetically controlled memristor from model (1): (**a**) the *i*-*v* characteristic curve with frequency *f*; (**b**) the *i*-*v* characteristic curve with amplitude *A*.

**Figure 2 entropy-25-00203-f002:**
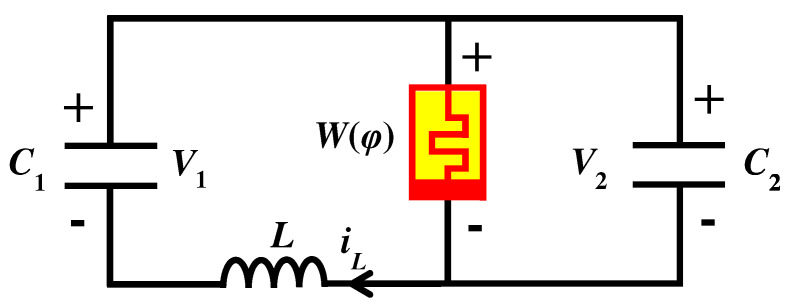
Structure of the simple memristor chaotic circuit.

**Figure 3 entropy-25-00203-f003:**
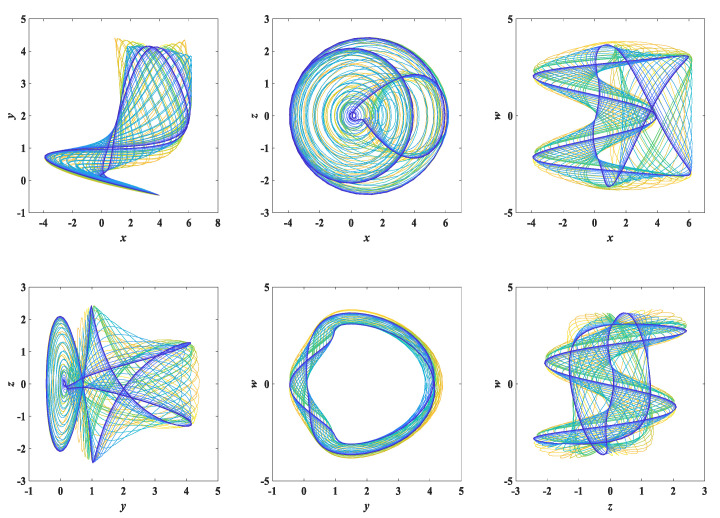
The state trajectory diagram of the simple memristor chaotic circuit model.

**Figure 4 entropy-25-00203-f004:**
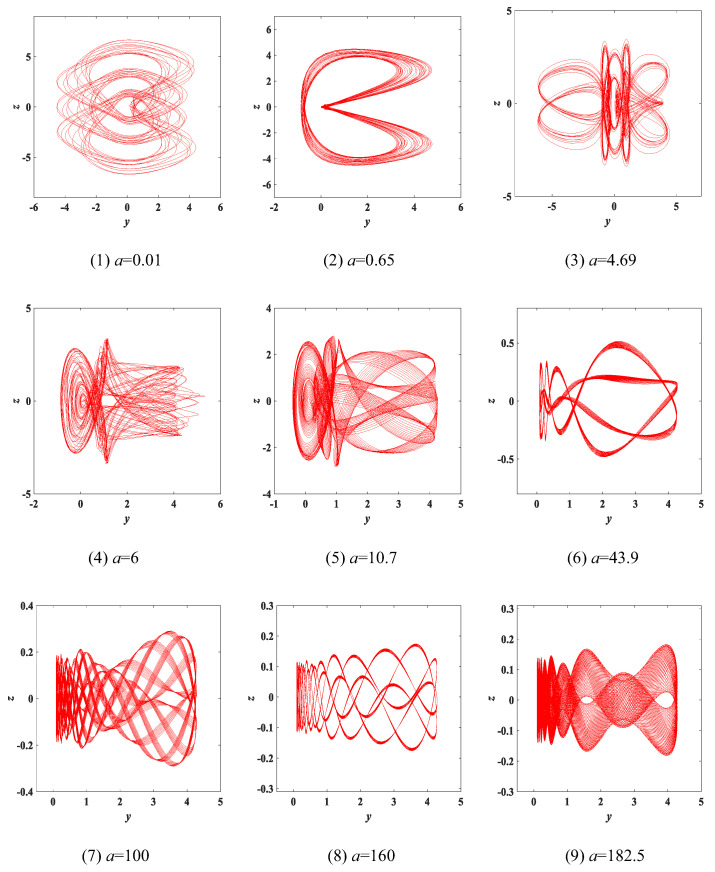
The type-1 to type-9 evolution diagram of the *y*-*z* chaotic attractors with varying parameter *a* of system (3), where *b* = 1 and *c* = 2.

**Figure 5 entropy-25-00203-f005:**
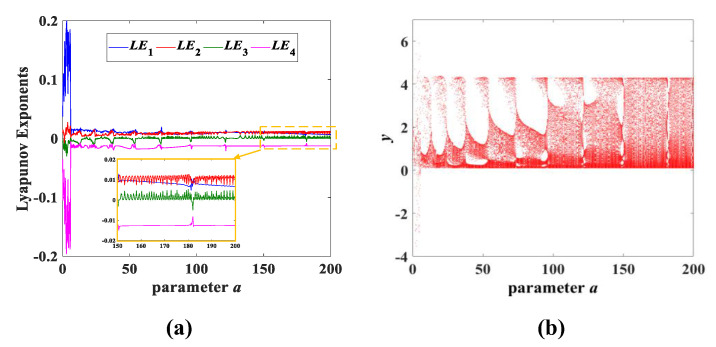
Lyapunov exponent spectrum and bifurcation diagram with parameter *a*: (**a**) Lyapunov exponent spectrum, where an enlarged portion of the yellow dotted box is shown at the bottom of subfigure (**a**); (**b**) bifurcation diagram.

**Figure 6 entropy-25-00203-f006:**
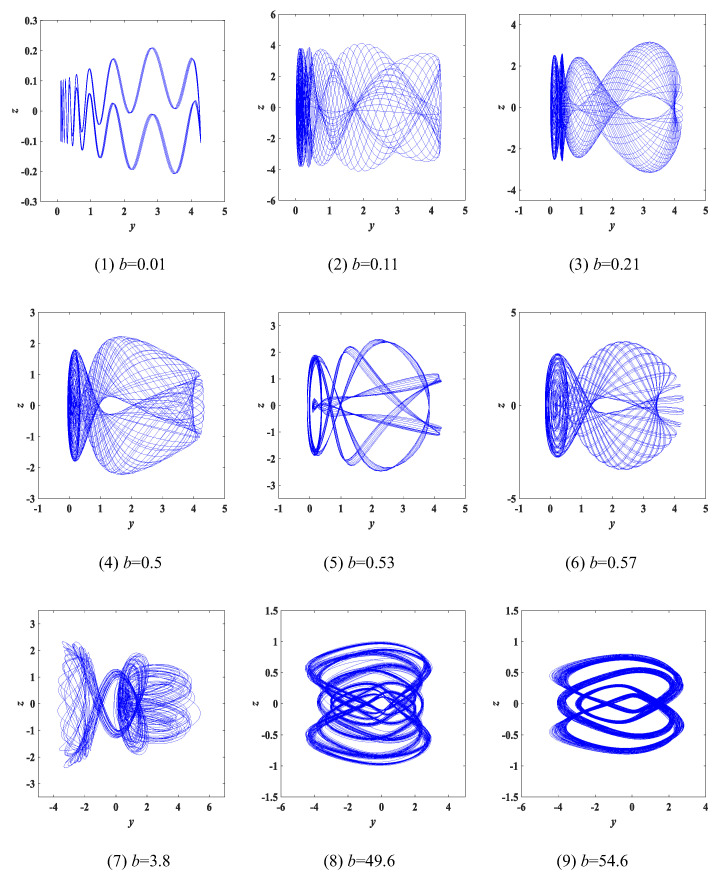
The type-1 to type-9 evolution diagrams of the *y*-*z* chaotic attractors with varying parameter *b* of system (3), where *a* = 8 and *c* = 2.

**Figure 7 entropy-25-00203-f007:**
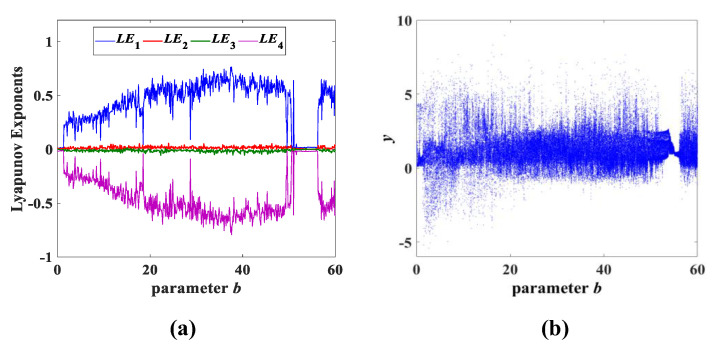
Lyapunov exponent spectrum and bifurcation diagram with parameter *b*: (**a**) Lyapunov exponent spectrum; (**b**) bifurcation diagram.

**Figure 8 entropy-25-00203-f008:**
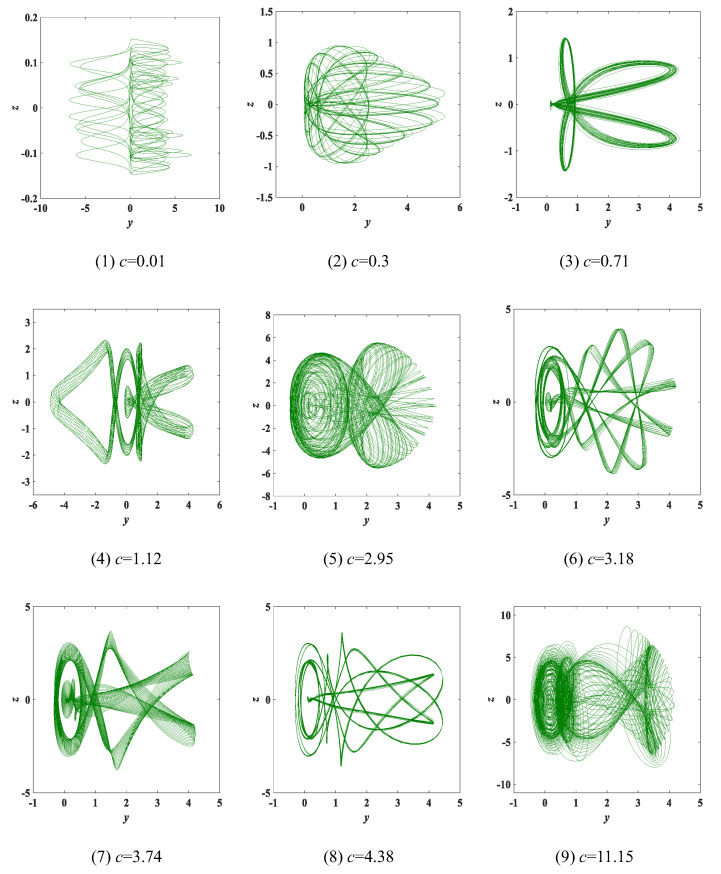
The type-1 to type-9 evolution diagrams of the *y*-*z* chaotic attractors with varying parameter *c* of system (3), where *a* = 8 and *b* = 1.

**Figure 9 entropy-25-00203-f009:**
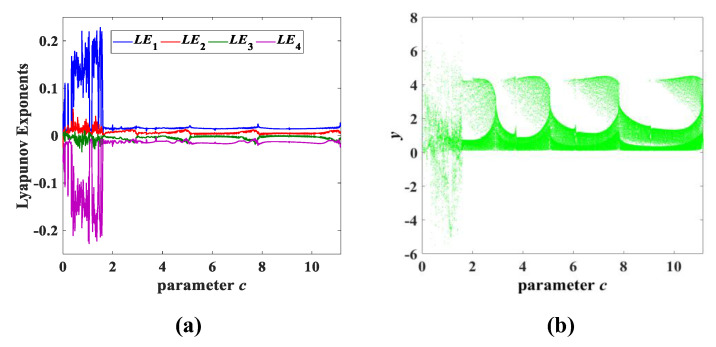
Lyapunov exponent spectrum and bifurcation diagram with parameter *c*: (**a**) Lyapunov exponent spectrum; (**b**) bifurcation diagram.

**Figure 10 entropy-25-00203-f010:**
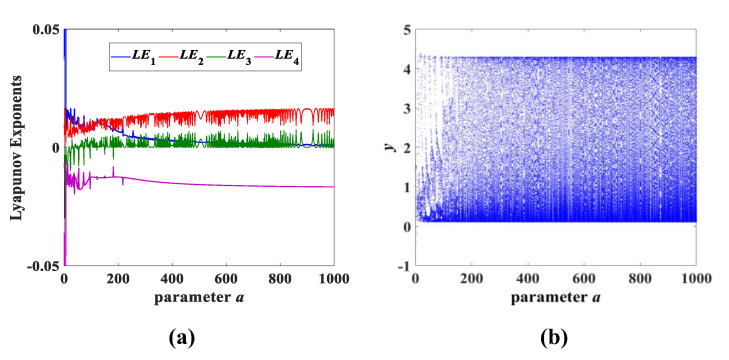
Lyapunov exponent spectrum and bifurcation diagram for parameter *a* variation while *b* = 1 and *c* = 2: (**a**) Lyapunov exponent spectrum; (**b**) bifurcation diagram.

**Figure 11 entropy-25-00203-f011:**
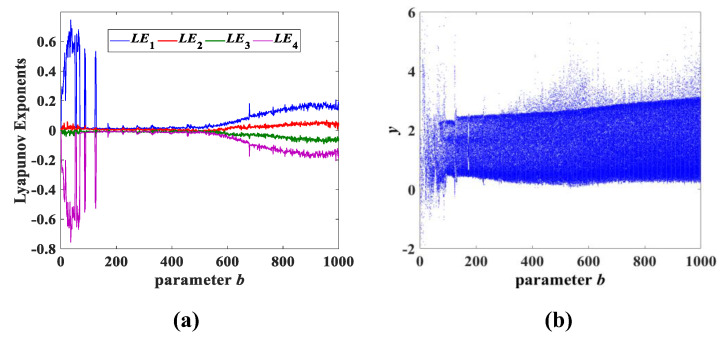
Lyapunov exponent spectrum and bifurcation diagram for parameter *b* variation while *a* = 8 and *c* = 2: (**a**) Lyapunov exponent spectrum; (**b**) bifurcation diagram.

**Figure 12 entropy-25-00203-f012:**
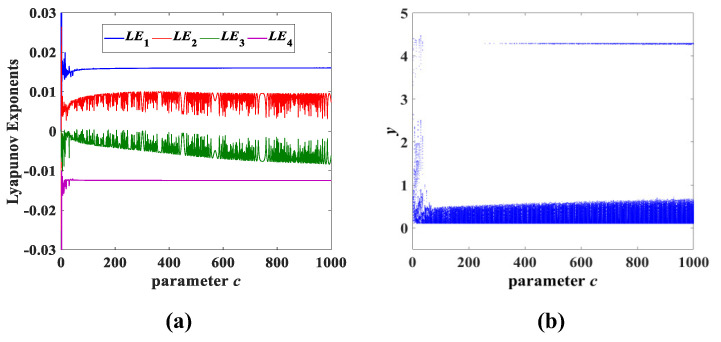
Lyapunov exponent spectrum and bifurcation diagram for parameter *c* variation while *a* = 8 and *b* = 1: (**a**) Lyapunov exponent spectrum; (**b**) bifurcation diagram.

**Figure 13 entropy-25-00203-f013:**
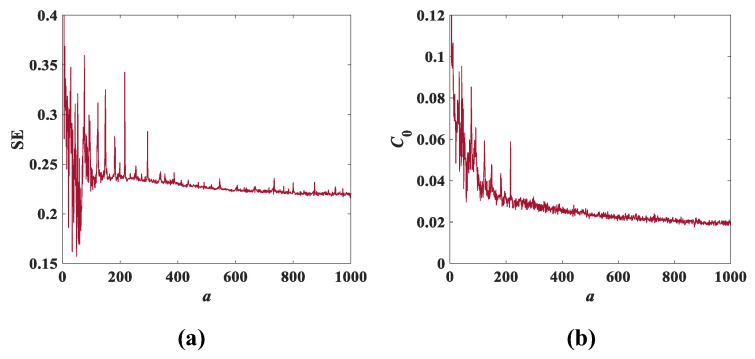
Complexity with parameter *a* when *b* = 1 and *c* = 2: (**a**) SE complexity; (**b**) $C_{0}$ complexity.

**Figure 14 entropy-25-00203-f014:**
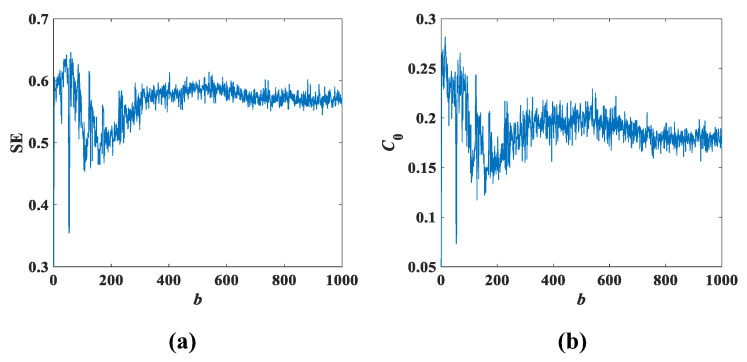
Complexity with parameter *b* when *a* = 8 and *c* = 2: (**a**) SE complexity; (**b**) $C_{0}$ complexity.

**Figure 15 entropy-25-00203-f015:**
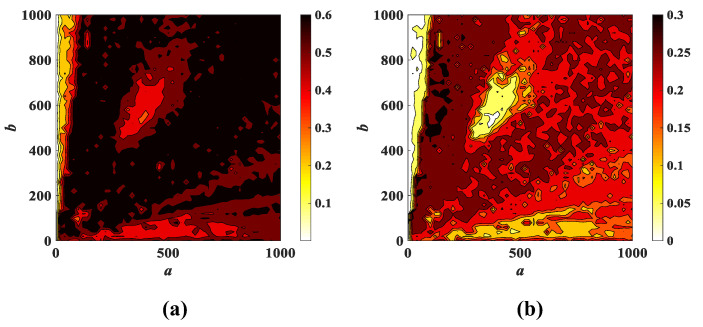
Complexity with parameters *a* and *b* when *c* = 2: (**a**) SE complexity; (**b**) $C_{0}$ complexity.

**Figure 16 entropy-25-00203-f016:**
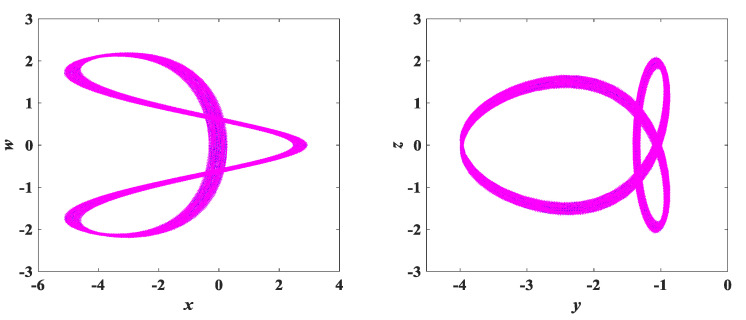
Coexisting attractor diagram with initial values of (0.1, −4, 0.1, 0.01/−0.01).

**Figure 17 entropy-25-00203-f017:**
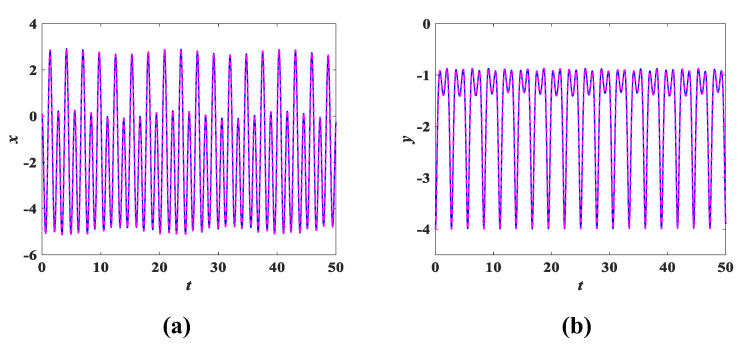
Time domain waveforms of coexisting attractors corresponding to initial values of (0.1, −4, 0.1, 0.01/−0.01): (**a**) *x* time-domain waveform; (**b**) *y* time-domain waveform.

**Figure 18 entropy-25-00203-f018:**
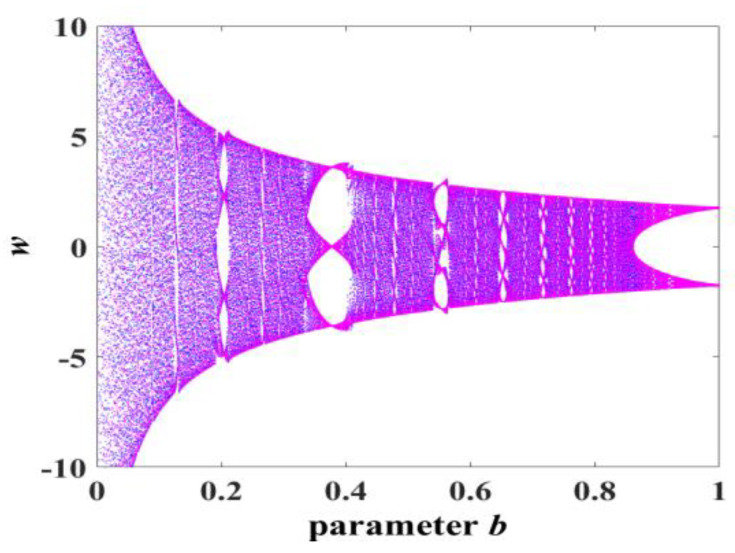
Bifurcation diagram of coexisting attractors corresponding to initial values of (0.1, −4, 0.1, 0.01/−0.01).

**Figure 19 entropy-25-00203-f019:**
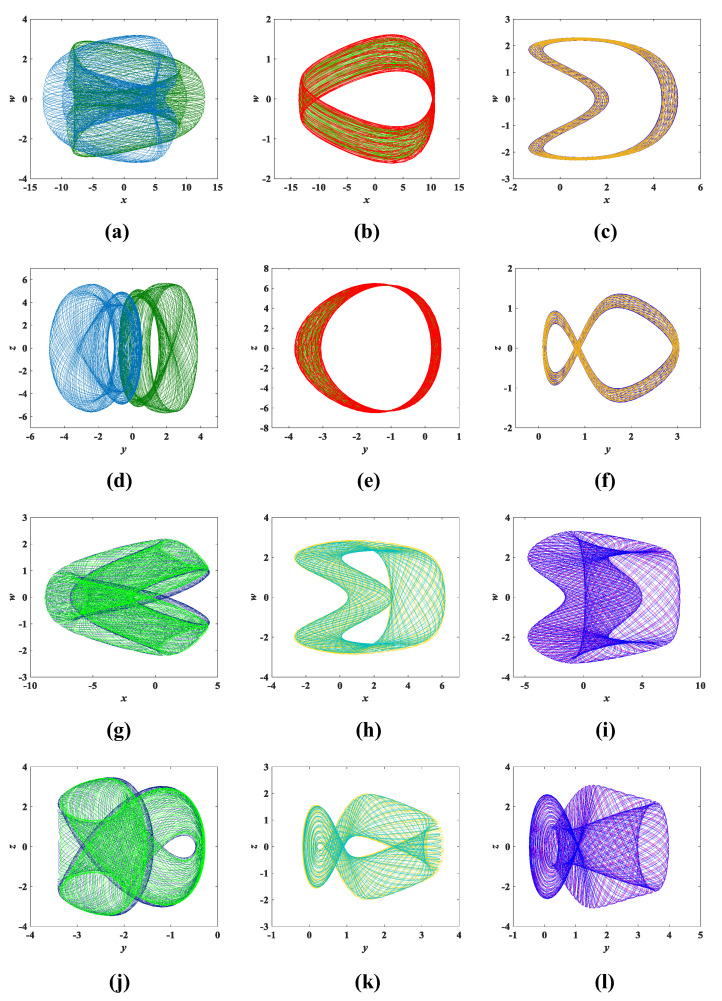
Coexistence attractor diagram corresponding to different initial conditions: (**a**) *x*-*w* coexistence attractor trajectory with initial condition (13/−13, 1/−1, 0.1, 0.1). (**b**) *x*-*w* coexistence attractor trajectory with initial condition (−12, 0.1, 3/−3, 0.1). (**c**) *x*-*w* coexistence attractor trajectory with initial condition (2, 0.1, 0.1, 0.01/−0.01). (**d**) *y*-*z* coexistence attractor trajectory with initial condition (13/−13, 1/−1, 0.1, 0.1). (**e**) *y*-*z* coexistence attractor trajectory with initial condition (−12, 0.1, 3/−3, 0.1). (**f**) *x*-*w* coexistence attractor trajectory with initial condition (2, 0.1, 0.1, 0.01/−0.01). (**g**) *x*-*w* coexistence attractor trajectory with initial condition (0.1, −3, 3/−3, 0.1). (**h**) *x*-*w* coexistence attractor trajectory with initial condition (0.1, 0.5, 0.1, 2/−2). (**i**) *x*-*w* coexistence attractor trajectory with initial condition (0.1, 0.1, 2/−2, −2/2). (**j**) *y*-*z* coexistence attractor trajectory with initial condition (0.1, −3, 3/−3, 0.1). (**k**) *y*-*z* coexistence attractor trajectory with initial condition (0.1, 0.5, 0.1, 2/−2). (**l**) *y*-*z* coexistence attractor trajectory with initial condition (0.1, 0.1, 2/−2, −2/2).

**Figure 20 entropy-25-00203-f020:**
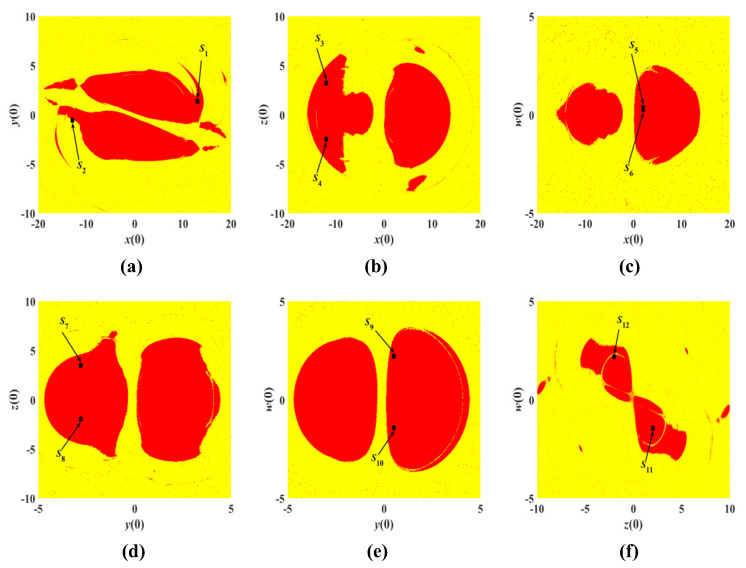
Basins of attraction with any two initial values: (**a**) *x*(0)-*y*(0) plane. (**b**) *x*(0)-*z*(0) plane. (**c**) *x*(0)-*w*(0) plane. (**d**) *y*(0)-*z*(0) plane. (**e**) *y*(0)-*w*(0) plane. (**f**) *z*(0)-*w*(0) plane.

**Figure 21 entropy-25-00203-f021:**
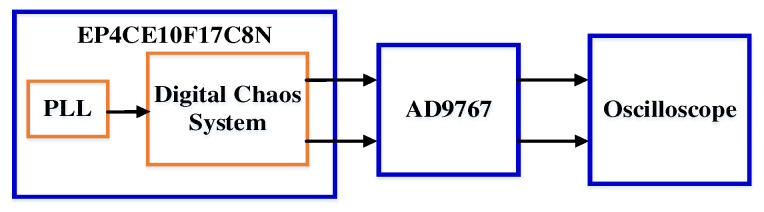
Experimental hardware platform plot.

**Figure 22 entropy-25-00203-f022:**
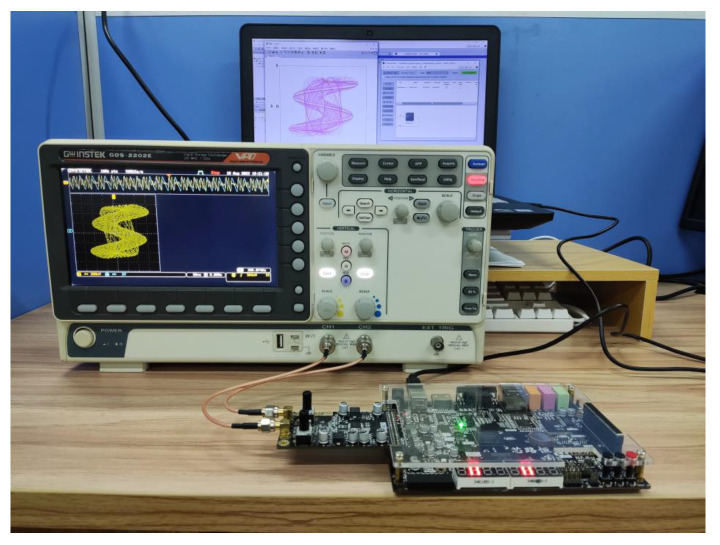
FPGA experimental platform connection diagram, which shows the comparison of the *z*-*w* trajectory with an oscilloscope and computer.

**Figure 23 entropy-25-00203-f023:**
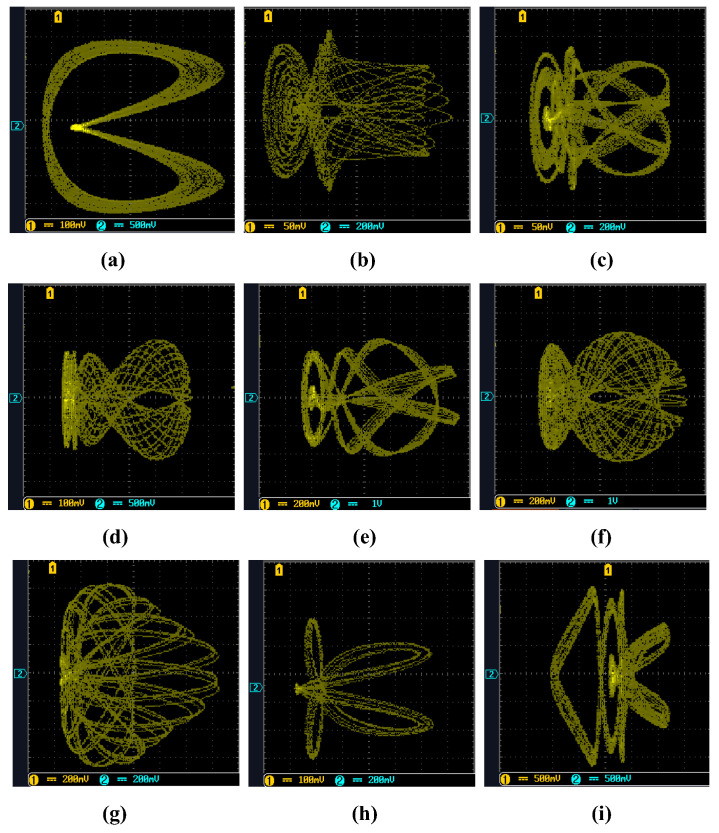
FPGA implementation of *y*-*z* chaotic attractors with different parameters (a,b,c): (**a**) the attractor trajectory corresponding to *a* = 0.65; (**b**) the attractor trajectory corresponding to *a* = 6.00; (**c**) the attractor trajectory corresponding to *a* = 10.70; (**d**) the attractor trajectory corresponding to *b* = 0.21; (**e**) the attractor trajectory corresponding to *b* = 0.53; (**f**) the attractor trajectory corresponding to *b* = 0.57; (**g**) the attractor trajectory corresponding to *c* = 0.30; (**h**) the attractor trajectory corresponding to *c* = 0.71; (**i**) the attractor trajectory corresponding to *c* = 1.12.

**Table 1 entropy-25-00203-t001:** Relevant information of each type of *y*-*z* chaotic attractor corresponding to different parameter *a* values.

No.	Parameter *a*	Lyapunov Exponents	State	Lyapunov Dimension
1	0.01	0.0609, 0.0144, 0, −0.0529	Hyperchaos	4.4234
2	0.65	0.0156, 0, 0, −0.0112	Chaos	4.3929
3	4.69	0.1745, 0, −0.0118, −0.1566	Chaos	4.0389
4	6.00	0.0388, 0.0125, 0, −0.0440	Hyperchaos	4.1659
5	10.70	0.0123, 0.0115, 0, −0.0144	Hyperchaos	4.6528
6	43.90	0.0140, 0.0078, 0, −0.0133	Hyperchaos	4.6391
7	100.00	0.0112, 0.0108, 0, −0.0128	Hyperchaos	4.7188
8	160.00	0.0112, 0.0092, 0, −0.0127	Hyperchaos	4.6063
9	182.50	0.0108, 0.0091, 0, −0.0127	Hyperchaos	4.5669

**Table 2 entropy-25-00203-t002:** Relevant information of each type of *y*-*z* chaotic attractor corresponding to different parameter *b* values.

No.	Parameter *b*	Lyapunov Exponents	State	Lyapunov Dimension
1	0.01	0.0152, 0, 0, −0.0174	Chaos	3.8736
2	0.11	0.0103, 0.0100, 0, −0.0134	Hyperchaos	4.5149
3	0.21	0.0111, 0.0086, 0, −0.0139	Hyperchaos	4.4173
4	0.50	0.0139, 0.0058, 0, −0.0115	Hyperchaos	4.7130
5	0.53	0.0151, 0.0053, 0, −0.0122	Hyperchaos	4.6721
6	0.57	0.0157, 0.0076, 0, −0.0124	Hyperchaos	4.8790
7	3.80	0.1138, 0, −0.0143, −0.1006	Chaos	3.9891
8	49.60	0.0142, 0, 0, −0.0161	Chaos	3.8820
9	54.60	0.0172, 0, 0, −0.0187	Chaos	3.9198

**Table 3 entropy-25-00203-t003:** Relevant information of each type of *y*-*z* chaotic attractor corresponding to different parameter *c* values.

No.	Parameter *c*	Lyapunov Exponents	State	Lyapunov Dimension
1	0.01	0.0505, 0.0080, 0, −0.0575	Hyperchaos	4.0174
2	0.30	0.0156, 0.0118, 0, −0.0163	Hyperchaos	4.6810
3	0.71	0.1225, 0.0124, 0, −0.1212	Hyperchaos	4.1130
4	1.12	0.0103, 0.0053, 0, −0.0115	Hyperchaos	4.3565
5	2.95	0.0171, 0, 0, −0.0130	Chaos	4.3154
6	3.18	0.0173, 0.0070, 0, −0.0125	Hyperchaos	4.9440
7	3.74	0.0166, 0, −0.0077, −0.0122	Chaos	3.7295
8	4.38	0.0159, 0, 0, −0.0156	Chaos	4.0192
9	11.15	0.0192, 0, 0, −0.0184	Chaos	4.0435

**Table 4 entropy-25-00203-t004:** Lyapunov exponents, Lyapunov dimension and dynamic behavior with parameters *a* and *b*.

Parameter *a*	Parameter *b*	Lyapunov Exponents	Lyapunov Dimension	Dynamic Behavior
8	1	0.0433, 0.0159, 0, −0.0521	4.1363	hyperchaotic state
200	1	0.0495, 0.0079, 0, −0.0366	4.5683	hyperchaotic state
8	100	0.0478, 0, 0, −0.0524	4.1363	chaotic state
1000	100	0.0617, 0, −0.0172, −0.0501	3.8882	chaotic state

**Table 5 entropy-25-00203-t005:** Coexisting attractors corresponding to different initial conditions.

Changed Initial Variable Values	Initial Conditions	Figure 19
$x_{0}$, $y_{0}$	$S_{1}$ = (13, 1, 0.1, 0.1), $S_{2}$ = (−13, −1, 0.1, 0.1)	(a), (d)
$x_{0}$, $z_{0}$	$S_{3}$ = (−12, 0.1, 3, 0.1), $S_{4}$ = (−12, 0.1, −3, 0.1)	(b), (e)
$x_{0}$, $w_{0}$	$S_{5}$ = (2, 0.1, 0.1, 0.01), $S_{6}$ = (2, 0.1, 0.1, −0.01)	(c), (f)
$y_{0}$, $z_{0}$	$S_{7}$ = (0.1, −3, 3, 0.1), $S_{8}$ = (0.1, −3, −3, 0.1)	(g), (j)
$y_{0}$, $w_{0}$	$S_{9}$ = (0.1, 0.5, 0.1, 2), $S_{10}$ = (0.1, 0.5, 0.1, −2)	(h), (k)
$z_{0}$, $w_{0}$	$S_{11}$ = (0.1, 0.1, 2, −2), $S_{12}$ = (0.1, 0.1, −2, 2)	(i), (l)

## Data Availability

Not applicable.

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
