# Peer review of "Implementation of the Simple Hyperchaotic Memristor Circuit with Attractor Evolution and Large-Scale Parameter Permission"

_entropy, 2023, doi:10.3390/e25020203_

Round 1

Reviewer 1 Report

This is a carefully done study about a novel simple four-dimensional memristor circuit and the findings of its rich dynamical behaviors are of considerable detail. For the benefit of the reader, however, a few points need clarifying and certain statements require further justification. My detailed comments are as follows:

1.     This paper put forward a hyperchaotic memristor circuit and many findings are listed to show us its chaotic and beautiful hyperchaotic property, but there are no clear words in the keywords part to point out chaos or hyperchaos.

2.     For the introduction part you need to strengthen the logical connection of the content. You introduced the memristor and then hyperchaotic systems, but for the beginning of the second paragraph, you back to simple chaotic systems without showing the relationship with the system you proposed. You may want to note your system is hyperchaotic as well as simple, but the expression needs to be improved.

3.     It is noted that your contributions need to simplify. Points (2), (3), and (4) describe the different kinds of circuit properties but are they all special, representative or innovative? If the answer is yes, give your reasons. Hope you can think carefully and then make a clearer and more concise statement.

4.     What does attractor coexistence have to do with multiple stability? The attractor coexistence is explained in detail, even part 3.5 seemed to confirm the results in part 3.4. ( The conclusion is achieved as” Therefore, system (3) has multiple attractor coexistence behavior.”) However, in the abstract, the expression shows“ Then, the results from the basin of attraction reflect the chaotic property of the circuit as well as the multistability.” 

5.     Notice that the horizontal and vertical axes of many figures are the Y-axis and the z-axis. Why choose the y-z axis only?

6.     There are a number of circuit characteristics and dynamic behaviors analyses to describe the proposed circuit. Why did you choose these indexes? You are expected to give the reasons and the advantages of the conclusion, for instance, it can lead to some applications.

7.     For some formulas and definitions, such as the relative power spectrum probability, the spectral entropy, the normalized spectral entropy, and so on, It is recommended that you streamline your process and identify your sources.

8.     Check some typos and English representation. There are some obvious problems you should pay extra attention to. First, the sentence to explain the parameters in the system proposed above should be written at the beginning of the line without space. (sentences begin with “where” in part 2) Then, too many “, respectively” are used in this paper which caused repeatability of language. There are also some sentences subjects and objects that are too long to read and have some grammar problems, for example “According to Kirchhoff’s theorem, the relationship between the circuit elements and the magnetic-controlled memristor model (1), the dynamical behavior of the circuit is described as follows”.

Author Response

Dear Editor and reviewers,

Thank you for allowing a major revision of our manuscript, with an opportunity to address the reviewers’ comments.

We are uploading (a) our point-by-point response to the comments (below) (response to reviewers), (b) an updated manuscript highlighted in a different color.

We hope that our revised version will be satisfactory for acceptance. Great thank for you and reviewers’ hard work.

Best regards,

Gang Yang, Xiaohong Zhang*, Ata Jahangir Moshayedi

Reviewer 2 Report

- The manuscript must be reorganized in a more concise and standard form.

- The introduction (which should include a history of the theme) should end at line 60 where the own contributions begins.

- Some calculation details can also be removed or briefly presented.

- It is advisable to use some quantitative entropic descriptions (possibly at the end as synthesis parameters), not just qualitative ones such as spectral entropy complexity etc.

Author Response

(The authors gave the same response as above.)

Round 2

Reviewer 1 Report

Since all the comments have been addressed. The revised version can be accepted now.

Author Response

Many thanks to the reviewer for his (her) recognition and selfless guidance! We have also discovered the flaws in our research through the revision of this manuscript, and we will definitely continue to work hard to write papers with more theoretical value and application prospects in the future.

Reviewer 2 Report

...

Author Response

  1. Q: Does the introduction provide sufficient background and include all relevant references?

A: Thanks for the expert’s professional guidance. First, we have adapted the introduction of simple memristor circuits in the introduction to precede the introduction of hyperchaotic memristor circuits. (on page 1, marked with green color) Second, we have added a description of the relationship of our proposed system to the content of the introduction. (on page 2, marked with green color) Third, we have adjusted the contribution content to the position suggested by the experts and have simplified the contribution content appropriately. (on page 2, marked with red color, start at line 63)

  1. Q: Are the results clearly presented?

A: The expert’s consideration is greatly valuable for us. According to the last expert's suggestion of adding some quantitative entropy description, we have added the description of the average value of spectral entropy in subsection 3.3, and summarized and analyzed the results. (on page 17, marked with blue color).
